SCIENCE FORUM

# SARS-CoV-2 (COVID-19) by the numbers

**Abstract** The COVID-19 pandemic is a harsh reminder of the fact that, whether in a single human host or a wave of infection across continents, viral dynamics is often a story about the numbers. In this article we provide a one-stop, curated graphical source for the key numbers (based mostly on the peer-reviewed literature) about the SARS-CoV-2 virus that is responsible for the pandemic. The discussion is framed around two broad themes: i) the biology of the virus itself; ii) the characteristics of the infection of a single human host.

**YINON M BAR-ON, AVI FLAMHOLZ, ROB PHILLIPS AND RON MILO\***

**\*For correspondence:** ron.milo@ weizmann.ac.il

**Competing interests:** The authors declare that no competing interests exist.

## Introduction

The COVID-19 pandemic has made brutally clear the need for further research into many aspects of viruses. In this article we compile data about the basic properties of the SARS-CoV-2 virus, and about how it interacts with the body (*Figure 1*). We also discuss a number of questions about the virus, and perform 'back-of-the-envelope' calculations to show the insights that can be gained from knowing some key numbers and using quantitative reasoning. It is important to note that much uncertainty remains, and while 'back-of-the-envelope' calculations can improve our intuition through sanity checks, they cannot replace detailed epidemiological analysis.

## Eight questions about SARS-CoV-2

### 1. How long does it take a single infected person to yield one million infected people?

If everybody continued to behave as usual, how long would it take the pandemic to spread from one person to a million infected victims? The basic reproduction number, $R_0$, suggests each infection directly generates 2–4 more infections in the absence of countermeasures like physical distancing. Once a person is infected, it takes a period of time known as the 'latent period' before they are able to transmit the virus. The current best-estimate of the median latent time is $\approx 3$ days followed by $\approx 4$ days of close to maximal infectiousness (*Li et al., 2020a*; *He et al., 2020*). The exact durations vary among people, and some are infectious for much longer. Using $R_0 \approx 4$, the number of cases will quadruple every $\approx 7$ days or double every $\approx 3$ days. 1000-fold growth (going from one case to $10^3$) requires 10 doublings since $2^{10} \approx 10^3$; 3 days $\times$ 10 doublings = 30 days, or about one month. So we expect $\approx 1000$x growth in one month, a million-fold ($10^6$) in two months, and a billion fold ($10^9$) in three months. Even though this calculation is highly simplified, ignoring the effects of 'super-spreaders', herd-immunity and incomplete testing, it emphasizes the fact that viruses can spread at a bewildering pace when no countermeasures are taken. This illustrates why it is crucial to limit the spread of the virus by physical distancing measures. For fuller discussion of the meaning of $R_0$, the latent and infectious periods, as well as various caveats, see the section on 'Definitions and measurement methods' below.

### 2. What is the effect of physical distancing?

A highly simplified quantitative example helps clarify the need for physical distancing. Suppose that you are infected and you encounter 50 people over the course of a day of working, commuting, socializing and running errands. To make the numbers round, let's further suppose that you have a 2% chance of transmitting the virus in each of these encounters, so that you

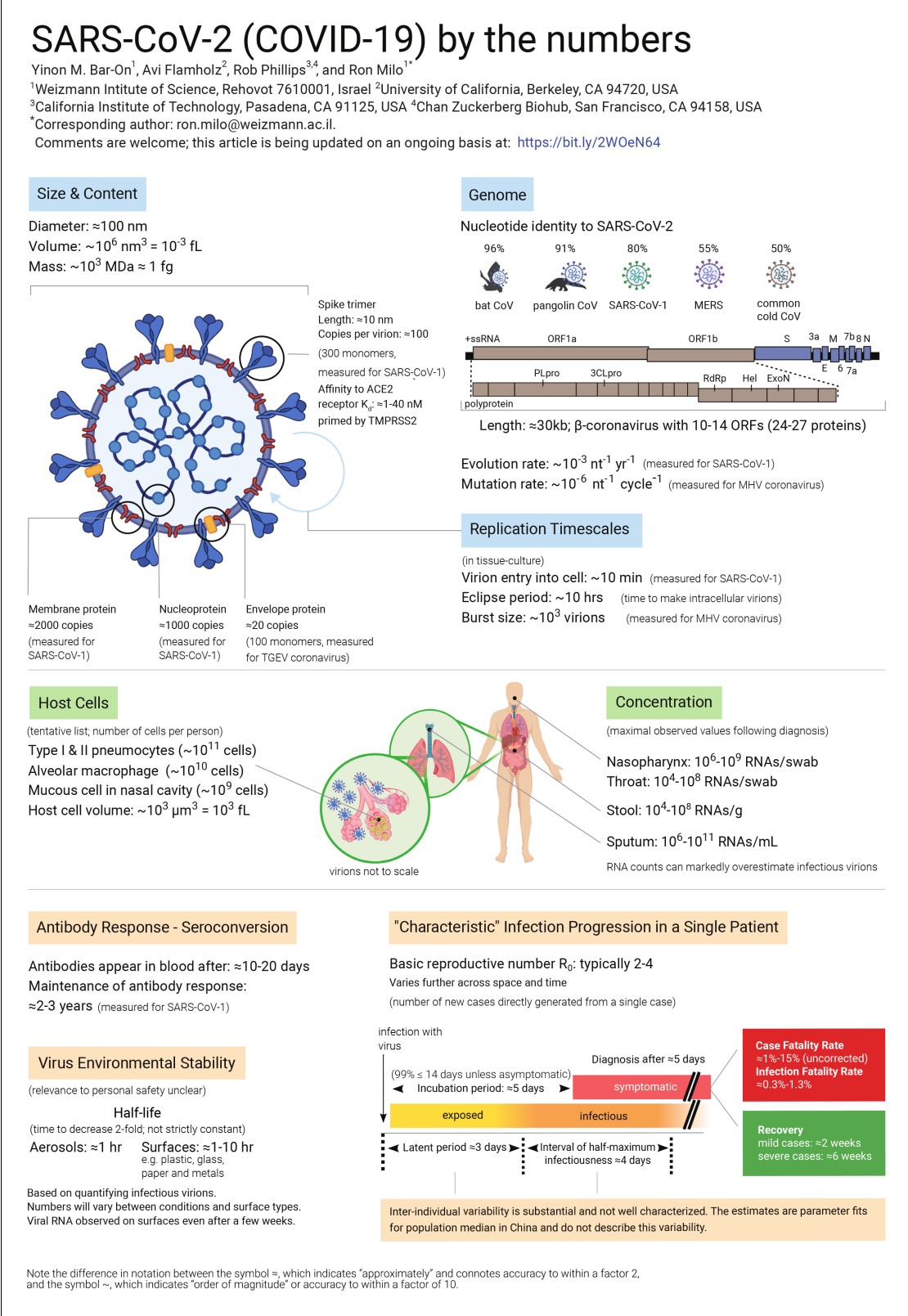

**Figure 1.** SARS-CoV-2 (COVID-19) by the numbers. Graphic showing what we know about the basic properties of the SARS-CoV-2 virus, such as its size and genome, and about how it interacts with the body. These topics are discussed further in the text, which also includes sources for all the values listed. This article will be updated as new data become available, and the latest version is available at: bit.ly/2WOeN64. A larger version of this figure (which was created with Biorender) is available as *Supplementary file 1*.

are likely to infect one new person each day. If you are infectious for 4 days, then you will infect four others on average, which is on the high end of the $R_0$ values for SARS-CoV-2 in the absence of physical distancing. If you instead see five people each day (preferably fewer) because of physical distancing, then you will infect 0.1 people per day, or 0.4 people before you become less infectious. The desired effect of physical distancing is to make each current infection produce <1 new infections. An effective reproduction number ($R_e$) smaller than one will ensure the number of infections eventually dwindles. It is critically important to quickly achieve $R_e < 1$, which is substantially more achievable than pushing $R_e$ to near zero through public health measures.

### 3. Why was the initial quarantine period two weeks?

The period of time from infection to symptoms is termed the incubation period. The median SARS-CoV-2 incubation period is estimated to be roughly 5 days (*Lauer et al., 2020*). Yet there is much person-to-person variation. Approximately 99% of those showing symptoms will show them before day 14, which explains the two week confinement period. Importantly, this analysis neglects infected people who never show symptoms. Since asymptomatic people are not usually tested, it is still not clear how many such cases there are or how long asymptomatic people remain infectious for.

### 4. How do N95 masks block SARS-CoV-2?

N95 masks are designed to remove more than 95% of all particles that are at least 0.3 microns (μm) in diameter. In fact, measurements of the particle filtration efficiency of N95 masks show that they are capable of filtering ≈99.8% of particles with a diameter of ≈0.1 μm (*Rengasamy et al., 2017*). SARS-CoV-2 is an enveloped virus ≈0.1 μm in diameter, so N95 masks are capable of filtering most free virions, but they do more than that. How so? Viruses are often transmitted through respiratory droplets produced by coughing and sneezing. Respiratory droplets are usually divided into two size bins, large droplets (>5 μm in diameter) that fall rapidly to the ground and are thus transmitted only over short distances, and small droplets (≤5 μm in diameter). Small droplets can evaporate into 'droplet nuclei', remain suspended in air for significant periods of time and could be inhaled. Some viruses, such as measles, can be transmitted by droplet nuclei (*Tellier et al., 2019*). Larger droplets are also known to transmit viruses, usually by settling onto surfaces that are touched and transported by hands onto mucosal membranes such as the eyes, nose and mouth (*CDC, 2020*). The characteristic diameter of large droplets produced by sneezing is ~100 μm (*Han et al., 2013*), while the diameter of droplet nuclei produced by coughing is on the order of ~1 μm (*Yang et al., 2007*). At present, it is unclear whether surfaces or air are the dominant mode of SARS-CoV-2 transmission, but N95 masks should provide some protection against both (*Jefferson et al., 2009*; *Leung et al., 2020*).

### 5. How similar is SARS-CoV-2 to the common cold and flu viruses?

SARS-CoV-2 is a beta-coronavirus whose genome is a single ≈30 kb strand of RNA. The flu is caused by an entirely different family of RNA viruses called influenza viruses. Flu viruses have smaller genomes (≈14 kb) encoded in eight distinct strands of RNA, and they infect human cells in a different manner than coronaviruses. The 'common cold' is caused by a variety of viruses, including some coronaviruses and rhinoviruses. Cold-causing coronaviruses (e.g. OC43 and 229E strains) are quite similar to SARS-CoV-2 in genome length (within 10%) and gene content, but different from SARS-CoV-2 in sequence (≈50% nucleotide identity) and infection severity. One interesting facet of coronaviruses is that they have the largest genomes of any known RNA viruses (≈30 kb). These large genomes led researchers to suspect the presence of a 'proofreading mechanism' to reduce the mutation rate and stabilize the genome. Indeed, coronaviruses have a proofreading exonuclease called ExoN, which explains their low mutation rates (~$10^{-6}$ per site per cycle) in comparison to influenza (≈3 × $10^{-5}$ per site per cycle; *Sanjuán et al., 2010*). This relatively low mutation rate will be of interest for future studies predicting the speed with which coronaviruses can evade our immunization efforts.

### 6. How much is known about the SARS-CoV-2 genome and proteome?

SARS-CoV-2 has a single-stranded positive-sense RNA genome that codes for 10 genes ultimately producing 26 proteins according to an NCBI annotation (NC_045512). How is it that 10 genes code for >20 proteins? One long gene, orf1ab, encodes a polyprotein that is cleaved into 16

## Box 1. Glossary

**Clinical measures**
**Incubation period**: time between exposure and symptoms.
**Seroconversion**: time between exposure to virus and detectable antibody response.

**Epidemiological inferences**
$R_0$: the average number of cases directly generated by an individual infection.
**Latent period:** time between exposure and becoming infective.
**Infectious period:** time for which an individual is infective.
**Interval of half-maximum infectiousness:** the time interval during which the probability of viral transmission is higher than half of the peak infectiousness. This interval is similar to the infectious period, but applies also in cases where the probability of infection is not uniform in time.

**Viral species**
**SARS-CoV-2**: Severe acute respiratory syndrome coronavirus 2. A β-coronavirus causing the present COVID-19 outbreak.
**SARS-CoV-1**: β-coronavirus that caused the 2002 SARS outbreak in China.
**MERS**: a β-coronavirus that caused the Middle East Respiratory Syndrome outbreak beginning in Jordan in 2012.
**MHV**: Murine hepatitis virus, a model β-coronavirus on which much laboratory research has been conducted.
**TGEV**: Transmissible gastroenteritis virus, a model α-coronavirus that infects pigs.
**229E and OC43:** two strains of coronavirus (α- and β- respectively) that cause a fraction of common colds.

**Viral life-cycle**
**Eclipse period:** time between viral entry and appearance of intracellular virions.
**Latent period (cellular level):** time between viral entry and appearance of extracellular virions. Not to be confused with the epidemiological latent period described above.
**Burst size**: the number of virions produced from infection of a single cell. More appropriately called 'per-cell viral yield' for non-lytic viruses like SARS-CoV-2.
**Virion**: a viral particle.
**Polyprotein**: a long protein that is proteolytically cleaved into a number of distinct proteins. Distinct from a polypeptide, which is a linear chain of amino acids making up a protein.

**Human biology**
**Alveolar macrophage:** immune cells found in the lung that engulf foreign material like dust and microbes ('professional phagocytes').
**Pneumocytes:** the non-immune cells in the lung.
$K_D$: apparent binding affinity. In this case, gives the concentration of spike protein needed for half-maximum binding of ACE2 receptor. $K_D$ is measured using surface chemistry approaches for membrane proteins such as ACE2.
**ACE2:** Angiotensin-converting enzyme 2, the mammalian cell surface receptor that SARS-CoV-2 binds.
**TMPRSS2:** Transmembrane protease, serine 2, a mammalian membrane-bound serine protease that cleaves the viral spike tri-mer after it binds ACE2, revealing a fusion peptide that participates in membrane fusion that enables subsequent injection of viral RNA into the host cytoplasm.
**Nasopharynx:** the space above the soft palate at the back of the nose that connects the nose to the mouth.

**Notation**
Note the difference in notation between the symbol ≈, which indicates 'approximately' and connotes accuracy to within a factor of 2, and the symbol ~, which indicates 'order of magnitude' or accuracy to within a factor of 10.

proteins by proteases that are themselves part of the polyprotein. In addition to proteases, the polyprotein encodes an RNA polymerase and associated factors to copy the genome, a proof-reading exonuclease, and several other non-structural proteins. The remaining genes predominantly code for structural components of the virus: i) the spike protein which binds the cognate receptor on a human or animal cell; ii) a nucleoprotein that packages the genome; iii) two membrane-bound proteins. Though much current work is centered on understanding the role of 'accessory' proteins in the viral life cycle, we estimate that it is currently possible to ascribe clear biochemical or structural functions to only about half of SARS-CoV-2 gene products.

## 7. What can we learn from the mutation rate of the virus?

Studying viral evolution, researchers commonly use two measures describing the rate of genomic change. The first is the evolutionary rate, which is defined as the average number of substitutions that become fixed per year in strains of the virus, given in units of mutations per site per year. The second is the mutation rate, which is the number of substitutions per site per replication cycle. How can we relate these two values? Consider a single site at the end of a year. The only measurement of a mutation rate in a β-coronavirus suggests that this site will accumulate $\sim 10^{-6}$ mutations in each round of replication. Each replication cycle takes $\sim 10$ hr, and so there are $10^3$ cycles/year. Multiplying the mutation rate by the number of replications, assuming neutrality and neglecting the effects of evolutionary selection, we arrive at $10^{-3}$ mutations per site per year, consistent with the evolutionary rate inferred from sequenced coronavirus genomes. As our estimate is consistent with the measured rate, we infer that the virus undergoes near-continuous replication in the wild, constantly generating new mutations that accumulate over the course of the year. Using our knowledge of the mutation rate, we can also draw inferences about single infections. For example, since the mutation rate is $\sim 10^{-6}$ mutations/site/cycle and an mL of sputum might contain upwards of $10^7$ viral RNAs, we infer that every site is mutated more than once in such samples.

## 8. How stable and infectious is the virion on surfaces?

To understand how SARS-CoV-2 can be transmitted, it is vitally important to characterize the stability of infectious virions on different types of surfaces like cardboard, plastics, and various metals. This is a very active area of current research. However, there are significant caveats associated with viral stability measurements. The measured stability depends on the quantity measured, for example, one can measure either infectious virions or viral RNA copies. The number of infectious virions is typically much lower than inferred from measurements of the viral genome (*Woelfel et al., 2020*). SARS-CoV-2 RNA has been detected on various surfaces several weeks after they were last touched (*Moriarty et al., 2020*), but infectiousness appears to degrade more quickly than RNA. When researchers measured the stability of infectious virions on surfaces, the numbers depended greatly on the type of surface and the medium carrying the virus, with the stability on plastic being much greater than on copper or steel, for example. Viral stability is also known to depend strongly on temperature and humidity (*Chin et al., 2020*). Therefore calculating the probability of human infection from exposure to contaminated surfaces is a complex task for which sufficient data is not yet available. As such, caution and protective measures should be taken. To gain some intuition for the importance of surface transmission, we consider an undiagnosed infectious person who touches surfaces tens of times during their infectious period. Prior to lockdown, these public surfaces will subsequently be touched by hundreds of other people. From the basic reproduction number $R_0 \approx 2$–$4$ we can infer that not everyone touching those surfaces will be infected. More detailed bounds on the risk of infection from touching surfaces urgently awaits study.

## Definitions and measurement methods

### What are the meanings of $R_0$, 'latent period' and 'infectious period'?

The basic reproduction number, $R_0$, estimates the average number of new infections directly generated by a single infectious person. The 0 subscript connotes that this refers to early stages of an epidemic, when everyone in the region is susceptible (that is, there is no immunity) and no countermeasures have been taken.

As geography and culture affect how many people we encounter daily, how much we touch them and share food with them, estimates of $R_0$ can vary between locales. Moreover, because $R_0$ is defined in the absence of countermeasures and immunity, we are usually only able to assess the effective R ($R_e$). At the beginning of an epidemic, before any countermeasures, $R_e \approx R_0$. Several days pass before a newly-infected person becomes infectious themselves. This 'latent period' is typically followed by several days of infectivity called the 'infectious period'.

It is important to understand that reported values for all these parameters are population averages inferred from epidemiological models fit to counts of infected, symptomatic, and dying patients. Because testing is always incomplete and model fitting is imperfect, and data will vary between different locations, there is substantial uncertainty associated with reported values. Moreover, these median or average best-fit values do not describe person-to-person variation. For example, viral RNA was detectable in patients with moderate symptoms for more than one week after the onset of symptoms, and more than two weeks in patients with severe symptoms (*ECDC, 2020*). Though detectable RNA is not the same as active virus, this evidence calls for caution in using uncertain, average parameters to describe a pandemic. Why have detailed distributions of these parameters across people not been published? Direct measurement of latent and infectious periods at the individual level is extremely challenging, as accurately identifying the precise time of infection is usually very difficult.

### What is the difference between measurements of viral RNA and infectious viruses?

Diagnosis and quantification of viruses utilizes several different methodologies. One common approach is to quantify the amount of viral RNA in an environmental (e.g., surface) or clinical (e.g., sputum) sample via quantitative reverse-transcription polymerase chain reaction (RT-qPCR). This method measures the number of copies of viral RNA in a sample. The presence of viral RNA does not necessarily imply the presence of infectious virions. Virions could be defective (e.g., by mutation) or might have been deactivated by environmental conditions. To assess the concentration of infectious viruses, researchers typically measure the '50% tissue-culture infectious dose' ($TCID_{50}$). Measuring $TCID_{50}$ involves infecting replicate cultures of susceptible cells with dilutions of the virus and noting the dilution at which half the replicate dishes become infected. Viral counts reported by $TCID_{50}$ tend to be much lower than RT-qPCR measurements, which could be one reason why studies relying on RNA measurements (*Moriarty et al., 2020*) report the persistence of viral RNA on surfaces for much longer times than studies relying on $TCID_{50}$ (*van Doremalen et al., 2020*). It is important to keep this caveat in mind when interpreting data about viral loads, for example a report measuring viral RNA in patient stool samples for several days after recovery (*Wu et al., 2020a*). Nevertheless, for many viruses even a small dose of virions can lead to infection. For the common cold, for example, ~0.1 $TCID_{50}$ are sufficient to infect half of the people exposed (*Couch et al., 1966*).

### What is the difference between the case fatality rate and the infection fatality rate?

Global statistics on new infections and fatalities are pouring in from many countries, providing somewhat different views on the severity and progression of the pandemic. Assessing the severity of the pandemic is critical for policy making and thus much effort has been put into quantifying key measures of its progression. The most common measure for the severity of a disease is the fatality rate. One commonly reported measure is the case fatality rate (CFR), which is the proportion of fatalities out of total diagnosed cases. The CFR reported in different countries varies significantly, from 1% to about 15%. Several key factors affect the CFR. First, demographic parameters and practices associated with increased or decreased risk differ greatly across societies. For example, the prevalence of smoking, the average age of the population, and the capacity of the healthcare system. Indeed, the majority of people dying from SARS-CoV-2 have a preexisting condition such as cardiovascular disease or smoking (*The Novel Coronavirus Pneumonia Emergency Response Epidemiology Team, 2020*). There is also potential for bias in estimating the CFR. For example, a tendency to identify more severe cases (selection bias) will tend to overestimate the CFR. On the other hand, there is usually a delay between the onset of symptoms and death, which can lead to an underestimate of the CFR early in the progression of an epidemic. We report the uncorrected CFR values, and thus these caveats should be borne in mind. Even when correcting for these factors, the CFR does not give a complete picture as many cases with

mild or no symptoms are not tested. Thus, the CFR will tend to overestimate the rate of fatalities per infected person, termed the infection fatality rate (IFR). Estimating the total number of infected people is usually accomplished by testing a random sample for anti-viral antibodies, whose presence indicates that the patient was previously infected. At the time of writing, such assays are not widely available, and so researchers resort to surrogate datasets generated by testing of foreign citizens returning home from infected countries (*Verity et al., 2020*; *Nishiura et al., 2020*), large-scale semi-random testing in countries such as Iceland, near complete testing of passengers on the Diamond Princess ship (*Russell et al., 2020*), or epidemiological models estimating the number of undocumented cases (*Li et al., 2020a*; *Mizumoto et al., 2020*). These methods have their own caveats and uncertainties associated with them, and it is not entirely clear how representative they are but they do provide a first glimpse of the true severity of the disease.

### What is the burst size and the replication time of the virus?

Two important characteristics of the viral life cycle are the time it takes them to produce new infectious progeny, and the number of progeny each infected cell produces. The yield of new virions per infected cell is more clearly defined in lytic viruses, such as those infecting bacteria (bacteriophages), as viruses replicate within the cell and subsequently lyse the cell to release a 'burst' of progeny. This measure is usually termed 'burst size'. SARS-CoV-2 does not release its progeny by lysing the cell, but rather by continuous budding (*Park et al., 2020b*). Even though there is no 'burst', we can still estimate the average number of virions produced by a single infected cell. Measuring the time to complete a replication cycle or the burst size in vivo is very challenging, and thus researchers usually resort to measuring these values in tissue-culture. There are various ways to estimate these quantities, but a common and simple one is using 'one-step' growth dynamics. The key principle of this method is to ensure that only a single replication cycle occurs. This is typically achieved by infecting the cells with a large number of virions, such that every cell gets infected, thus leaving no opportunity for secondary infections.

Assuming entry of the virus to the cells is rapid (we estimate 10 min for SARS-CoV-2), the time it takes to produce progeny can be estimated by quantifying the lag between inoculation and the appearance of new intracellular virions, also known as the 'eclipse period'. This eclipse period does not account for the time it takes to release new virions from the cell. The time from cell entry until the appearance of the first extracellular viruses, known as the 'latent period' (not to be confused with the epidemiological latent period; see glossary in *Box 1*), estimates the duration of the full replication cycle. The burst size can be estimated by waiting until virion production saturates, and then dividing the total virion yield by the number of cells infected. While both the time to complete a replication cycle and the burst size may vary significantly in an animal host due to factors including the type of cell infected or the action of the immune system, these numbers provide us with an approximate quantitative view of the viral life-cycle at the cellular level.

### Are people usually diagnosed before or after they are contagious?

Our personal experience with infectious diseases leaves us with the intuition that we are contagious when we have symptoms. For the seasonal flu, for example, most transmissions indeed occur after a person has developed symptoms (*Ip et al., 2017*). For SARS-CoV-2, in contrast, it is common to be contagious before symptoms. The SARS-CoV-2 incubation period is about 5 days, while peak infectiousness begins two days before symptoms reveal themselves. As a result, a large fraction of infections occur pre-symptomatically, that is, without the infectious person realizing they have the disease (*Ferretti et al., 2020*; *He et al., 2020*). With testing capacity under strain, diagnosis typically occurs ≈5 days after symptom onset, or ≈10 days after infection. By that time, most people have already passed peak infectiousness. In order to effectively slow the growth of the pandemic, it is important to detect infections as early as possible and quarantine those who test positive. In the case of SARS-CoV-2 this means detection before symptoms because there is strong evidence of significant pre-symptomatic transmission. Finally, the situation is further complicated by a large fraction of asymptomatic cases, that is cases in which the infected person never develops noticeable symptoms. This fraction is more than half of children and young adults (*Davies et al., 2020*). Leading modeling efforts assume that asymptomatic infections are anywhere between 10–80% as contagious as symptomatic ones (*Ferretti et al., 2020*;

*Davies et al., 2020*). This wide range reflects a crucial gap in our understanding of SARS-CoV-2 transmission: great uncertainty about the magnitude of asymptomatic transmission.

## Sources of the numbers in Figure 1

Note that for about 10 out of 45 parameters, the literature values are from other coronaviruses. We await corresponding measurements for SARS-CoV-2.

### Size and content

**Diameter**. (Figure 3 in *Zhu et al., 2020*): "Electron micrographs of negative-stained 2019-nCoV particles were generally spherical with some pleomorphism. Diameter varied from about 60 to 140 nm."

**Volume**. Using diameter and assuming the virus is a sphere.

**Mass**. Using the volume and a density of ~1 g per mL.

**Number of spike trimers**. (*Neuman et al., 2011*): "Our model predicts ~90 spikes per particle."

**Length of spike trimers**. (*Zhu et al., 2020*): "Virus particles had quite distinctive spikes, about 9 to 12 nm, and gave virions the appearance of a solar corona."

**Receptor binding affinity** ($K_d$). *Walls et al., 2020* reports $K_d$ of $\approx 1$ nM for the binding domain using biolayer interferometry with $k_{on}$ of $\approx 1.5 \times 10^5$ $M^{-1}$ $s^{-1}$ and $k_{off}$ of $\approx 1.6 \times 10^{-4}$ $s^{-1}$ (Table 1). *Wrapp et al., 2020* reports $K_d$ of $\approx 15$ nM for the spike (Figure 3) and $\approx 35$ nM for the binding domain (Figure 4) using surface plasmon resonance with $k_{on}$ of $\approx 1.9 \times 10^5$ $M^{-1}$ $s^{-1}$ and $k_{off}$ of $\approx 2.8 \times 10^{-3}$ $s^{-1}$ for the spike, and $k_{on}$ of $\approx 1.4 \times 10^5$ $M^{-1}$ $s^{-1}$ and $k_{off}$ of $\approx 4.7 \times 10^{-3}$ $s^{-1}$ for the binding domain. *Lan et al., 2020* reports $K_d$ of $\approx 5$ nM for the binding domain (Extended Data Figure 4) using surface plasmon resonance with $k_{on}$ of $\approx 1.4 \times 10^6$ $M^{-1}$ $s^{-1}$ and $k_{off}$ of $\approx 6.5 \times 10^{-3}$ $s^{-1}$. *Shang et al., 2020* reports $K_d$ of $\approx 40$ nM for the binding domain (Extended Data Figure 6) using surface plasmon resonance with $k_{on}$ of $\approx 1.8 \times 10^6$ $M^{-1}$ $s^{-1}$ and $k_{off}$ of $\approx 7.8 \times 10^{-3}$ $s^{-1}$. The main disagreement between the studies seems to be on the $k_{off}$.

**Membrane** (M; 222 aa). (*Neuman et al., 2011*): "Using the M spacing data for each virus (Figure 6C), this would give ~1100 M2 molecules per average SARS-CoV, MHV and FCoV particle."

**Envelope** (E; 75 aa). (*Godet et al., 1992*): "Based on the estimated molar ratio and assuming that coronavirions bear 100 (*J Gen Virol* **63**: 241–245) to 200 spikes, each composed of 3 s molecules (*Virus Research* **20**:107–120) it can be inferred that approximately 15–30 copies of ORF4 protein are incorporated into TGEV virions (Purdue strain)."

**Nucleoprotein** (364 aa). (*Neuman et al., 2011*): "Estimated ratios of M to N protein in purified coronaviruses range from about 3M:1N (*Cavanagh, 1983*; *Escors et al., 2001*) to 1M:1N (*Hogue and Brian, 1986*; *Liu and Inglis, 1991*), giving 730–2200 N molecules per virion."

### Genome

**Type**. (ViralZone) +ssRNA "Monopartite, linear ssRNA(+) genome"

**Genome length**. The initial isolate of SARS-CoV-2 from Wuhan, China has a 29903 nt $\approx$ 30 kb ssRNA genome (NCBI MN908947.3), which is typical of a coronavirus (*Smith and Denison, 2012*).

(*Wu et al., 2020b*): "SARS-CoV-2 genome has 10 open reading frames (Figure 2A)". (*Wu et al., 2020c*): "The 2019-nCoV genome was annotated to possess 14 ORFs encoding 27 proteins". Coronavirus genomes contain several 'accessory proteins' that are not essential for replication and are not always expressed. The 'nonstructural proteins' are expressed as a polyprotein which is proteolytically cleaved into $\approx 10$ proteins. As transcription start and protease cleavage sites are not trivial to identify bioinformatically, there is some uncertainty about the exact number of transcriptional units and proteins expressed by SARS-CoV-2.

**Number of proteins**. (*Wu et al., 2020b*): "By aligning with the amino acid sequence of SARS PP1ab and analyzing the characteristics of restriction cleavage sites recognized by 3CLpro and PLpro, we speculated 14 specific proteolytic sites of 3CLpro and PLpro in SARS-CoV-2 PP1ab (Figure 2B). PLpro cleaves three sites at 181–182, 818–819, and 2763–2764 at the N-terminus and 3CLpro cuts at the other 11 sites at the C-terminus, and forming 15 non-structural proteins."

**Evolution rate**. (*Koyama et al., 2020*): "Mutation rates estimated for SARS, MERS, and OC43 show a large range, covering a span of 0.27 to 2.38 substitutions $\times$ 10–3/site/ year (see references 10–16)." Recent unpublished evidence also suggests this rate is of the same order of magnitude in SARS-CoV-2 ().

**Mutation rate**. (*Sanjuán et al., 2010*): "Murine hepatitis virus … Therefore, the corrected estimate of the mutation rate is $\mu_{s/n/c} = 1.9 \times 10^{-6} / 0.55 = 3.5 \times 10^{-6}$."

**Genome similarity**. For all species except pangolin, genomes were downloaded from NCBI and aligned to the SARS-CoV-2 reference (MN908947) with EMBOSS Stretcher (EMBL-EBI server). Reported values are percent nucleotide sequence identity. Genomes used: bat coronavirus RaTG13 (MN996532.1; 96% id); SARS-CoV-1 (NC_004718.3; 80% id); MERS (NC_019843.3; 55% id); human cold coronavirus strains OC43 (NC_006213.1; 53% id) and 229E (NC_002645.1; 50% id). For pangolin: '"PangolinCoV is 91.02% and 90.55% identical to SARS-CoV-2 and Bat-CoV RaTG13, respectively, at the whole genome level" (*Zhang et al., 2020*).

### Replication timescales

**Virion entry into cell** (**for SARS-CoV-1**). (*Schneider et al., 2012*): "Previous experiments had revealed that virus is internalized within 15 min". (*Ng et al., 2003*): "Within the first 10 min, some virus particles were internalized into vacuoles (arrow) that were just below the plasma membrane surface (Fig. 2, arrows). [...] The observation at 15 min postinfection (p.i.), did not differ much from 10 min p.i. (Fig. 4a)".

**Eclipse period**. (*Schneider et al., 2012*): "SARS-CoV replication cycle from adsorption to release of infectious progeny takes about 7 to 8 hr (data not shown)"; Figure 4 of *Harcourt et al., 2020* shows virions are released after 12–36 hr but because this is multi-step growth this represents an upper bound for the replication cycle.

**Burst size**. (*Hirano et al., 1976*): "The average per-cell yield of active virus was estimated to be about $6–7 \times 10^2$ plaque-forming units." This data is for MHV, so more research is needed to verify these values for SARS-CoV-2.

### Host cells

**Type**. (*Shieh et al., 2005*): "Immunohistochemical and in situ hybridization assays demonstrated evidence of SARS-associated coronavirus (SARS-CoV) infection in various respiratory epithelial cells, predominantly type II pneumocytes, and in alveolar macrophages in the lung". (*Walls et al., 2020*): "SARS-CoV-2 uses ACE2 to enter target cells". (*Rockx et al., 2020*): "In SARS-CoV-2-infected macaques, virus was excreted from nose and throat in absence of clinical signs, and detected in type I and II pneumocytes in foci of diffuse alveolar damage and mucous glands of the nasal cavity [...] In the upper respiratory tract, there was focal five or locally extensive SARS-CoV-2 antigen expression in epithelial cells of mucous glands in the nasal cavity (septum or concha) of all four macaques, without any associated histological lesions (fig. 2I)."

**Type I and Type II pneumocyte and alveolar macrophage cell number**. Values taken from table 4 in *Crapo et al., 1982*, and table 5 in *Stone et al., 1992*.

**Epithelial cells in mucous gland cell number and volume**. The value for the surface area of the nasal cavity is taken from *ICRP, 1975*; the value for the mucous gland density is taken from *Tos and Mogensen, 1976*; *Tos and Morgensen, 1977*; the value for the mucous gland volume is taken from *Widdicombe, 2019*; and the value for the mucous cell volume is taken from *Ordoñez et al., 2001* and *Mercer et al., 1994*. We divide the mucous gland volume by the mucous cell volume to arrive at the total number of mucous cells in a mucous gland. We multiply the surface density of mucous glands by the surface area of the nasal cavity to arrive at the total number of mucous glands, and then multiply the total number of mucous glands by the number of mucous cells per mucous gland.

**Type II pneumocyte volume**. (*Fehrenbach et al., 1995*): "Morphometry revealed that although inter-individual variation due to some oedematous swelling was present, the cells were in a normal size range as indicated by an estimated mean volume of $763 \pm 64 \ \mu m^3$."

**Alveolar macrophage volume**. (*Crapo et al., 1982*): "Alveolar macrophages were found to be the largest cell in the populations studied, having a mean volume of $2{,}491 \ \mu m^3$."

### Concentration

**Nasopharynx, throat, stool, and sputum**. We took the maximal viral load for each patient in nasopharyngeal swabs, throat swabs, stool or in sputum (figure 2 in *Wölfel et al., 2020*; figure 1 in *Kim et al., 2020*; *Pan et al., 2020*).

### Antibody response – seroconversion

**Seroconversion time** (**time period until a specific antibody becomes detectable in the blood**). (*Zhao et al., 2020*): "The seroconversion sequentially appeared for Ab, IgM and then IgG, with a median time of 11, 12 and 14 days, respectively". (*To et al., 2020*): "For 16 patients with serum samples available 14 days or longer after symptom onset, rates of seropositivity

were 94% for anti-NP IgG (n = 15), 88% for anti-NP IgM (n = 14), 100% for anti-RBD IgG (n = 16), and 94% for anti-RBD IgM (n = 15)".

**Maintenance of antibody response to virus**. (*Wu et al., 2007*): "Among 176 patients who had had severe acute respiratory syndrome (SARS), SARS-specific antibodies were maintained for an average of 2 years, and significant reduction of immunoglobulin G–positive percentage and titers occurred in the third year".

### Virus environmental stability

**Half-life on surfaces**. (*van Doremalen et al., 2020*): We use half-live values reported in Supplementary Table 1. *Chin et al., 2020*: We use short-term half-lives reported in the Appendix. *Pastorino et al., 2020*: We use the slopes of data poitns from the first two hours can calculate the short-term half-life from them. More studies are urgently needed to clarify the implications of virion stability on the probability of infection from aerosols or surfaces.

**RNA stability on surfaces** (*Moriarty et al., 2020*): "SARS-CoV-2 RNA was identified on a variety of surfaces in cabins of both symptomatic and asymptomatic infected passengers up to 17 days after cabins were vacated on the Diamond Princess but before disinfection procedures had been conducted (Takuya Yamagishi, National Institute of Infectious Diseases, personal communication, 2020)."

### 'Characteristic' infection progression in a single patient

**Basic reproductive number**, $R_0$. (*Li et al., 2020a*): "Our median estimate of the effective reproductive number, $R_e$ – equivalent to the basic reproductive number ($R_0$) at the beginning of the epidemic – is 2.38 (95% CI: 2.04–2.77)". (*Park et al., 2020a*): "Our estimated $R_0$ from the pooled distribution has a median of 2.9 (95% CI: 2.1–4.5)".

**Latent period** (from infection to being able to transmit). (*Li et al., 2020a*): "In addition, the median estimates for the latent and infectious periods are approximately 3.69 and 3.48 days, respectively"; see also table 1 in this paper. (*He et al., 2020*): We use the time it takes infectiousness to reach half its peak, which happens two days before symptom onset based on *Figure 1C*. As symptoms arise after five days (see 'Incubation period' below), this implies a three-day latent period.

**Incubation period** (from infection to symptoms). (*Lauer et al., 2020*): "The median

incubation period was estimated to be 5.1 days (95% CI, 4.5 to 5.8 days), and 97.5% of those who develop symptoms will do so within 11.5 days (CI, 8.2 to 15.6 days) of infection. These estimates imply that, under conservative assumptions, 101 out of every 10 000 cases (99th percentile, 482) will develop symptoms after 14 days of active monitoring or quarantine". (*Li et al., 2020b*): "The mean incubation period was 5.2 days (95% confidence interval [CI], 4.1 to 7.0), with the 95th percentile of the distribution at 12.5 days".

**Infectious period**. (*Li et al., 2020a*): "the median estimates for the latent and infectious periods are approximately 3.69 and 3.48 days, respectively"; see also table 1 in this paper. (*He et al., 2020*): We quantify the interval over which infectiousness is at least half its maximal value (the interval of half-maximal infectiousness) from the infectiousness profile in *Figure 1C*.

**Disease duration**. (*WHO, 2020*): "Using available preliminary data, the median time from onset to clinical recovery for mild cases is approximately 2 weeks and is 3–6 weeks for patients with severe or critical disease".

**Time until diagnosis**. (*Xu et al., 2020*): We used data on cases with known symptom onset and case confirmation dates and calculated the median time delay between these two dates.

**Case fatality rate**. (*ECDC, 2020*) - We use data from all countries with more than 50 death cases and calculate the uncorrected raw Case Fatality Rate for each country. The range represents the lowest and highest rates observed using ECDC data up to 14 April 2020.

**Infection fatality rate**. We rely on three independent approaches that estimate the IFR. The first relies on data about people who were extensively tested as a result of being repatriated. (*Verity et al., 2020*): "We obtain an overall IFR estimate for China of 0.66% (0.39%,1.33%)". (*Ferguson et al., 2020*): "The IFR estimates from Verity et al. have been adjusted to account for a non-uniform attack rate giving an overall IFR of 0.9% (95% credible interval 0.4–1.4%)". (*Nishiura et al., 2020*): "The infection fatality risk (IFR) – the actual risk of death among all infected individuals – is therefore 0.3% to 0.6%".

The second approach relies on data gathered from the Diamond Princess ship, where all passengers were tested. (*Russell et al., 2020*): "We estimated that the all-age cIFR on the Diamond Princess was 1.3% (95% confidence interval (CI): 0.38–3.6)".

The third approach relies on epidemiological modeling of case time-series from China. (*Mizumoto et al., 2020*): "We also found that most recent crude infection fatality ratio (IFR) and time-delay adjusted IFR is estimated to be 0.04% (95% CrI: 0.03–0.06%) and 0.12% (95% CrI: 0.08–0.17%)". Combining these three methods, and taking into account the reliability of each report, we estimate a crude range of ≈0.3–1.3% for the IFR.

## Acknowledgements

We thank the following individuals for productive feedback on this manuscript: Uri Alon, Niv Antonovsky, David Baltimore, Rachel Banks, Arren Bar Even, Naama Barkai, Molly Bassette, Menalu Berihoon, Biana Bernshtein, Pamela Bjorkman, Cecilia Blikstad, Julia Borden, Bill Burkholder, Griffin Chure, Lillian Cohn, Bernadeta Dadonaite, Emmie De wit, Ron Diskin, Ana Duarte, Tal Einav, Avigdor Eldar, Elizabeth Fischer, William Gelbart, Alon Gildoni, Britt Glausinger, Shmuel Gleizer, Dani Gluck, Soichi Hirokawa, Greg Huber, Christina Hueschen, Amit Huppert, Shalev Itzkovitz, Martin Jonikas, Leeat Keren, Gilmor Keshet, Marc Kirschner, Roy Kishony, Amy Kistler, Liad Levi, Sergei Maslov, Adi Millman, Amir Milo, Elad Noor, Gal Ofir, Alan Perelson, Steve Quake, Itai Raveh, Andrew Rennekamp, Tom Roeschinger, Daniel Rokhsar, Alex Rubinsteyn, Gabriel Salmon, Maya Schuldiner, Eran Segal, Ron Sender, Alex Sigal, Maya Shamir, Arik Shams, Mike Springer, Adi Stern, Noam Stern-Ginossar, Lubert Stryer, Dan Tawfik, Boris Veytsman, Aryeh Wides, Tali Wiesel, Anat Yarden, Yossi Yovel, Dudi Zeevi, Mushon Zer Aviv, and Alexander Zlokapa.

**Yinon M Bar-On** is in the Department of Plant and Environmental Sciences, Weizmann Institute of Science, Rehovot, Israel

https://orcid.org/0000-0001-8477-609X

**Avi Flamholz** is in the Department of Molecular and Cell Biology, University of California, Berkeley, Berkeley, United States

https://orcid.org/0000-0002-9278-5479

**Rob Phillips** is in the Department of Physics, Department of Applied Physics, and the Division of Biology and Biological Engineering, California Institute of Technology, Pasadena, and the Chan Zuckerberg Biohub, San Francisco, United States

https://orcid.org/0000-0003-3082-2809

**Ron Milo** is in the Department of Plant and Environmental Sciences, Weizmann Institute of Science, Rehovot, Israel

ron.milo@weizmann.ac.il

https://orcid.org/0000-0003-1641-2299

*Author contributions:* Yinon M Bar-On, Conceptualization, Resources, Data curation, Formal analysis, Validation, Investigation, Methodology, Writing - original draft, Writing - review and editing; Avi Flamholz, Resources, Data curation, Formal analysis, Validation, Investigation, Methodology, Writing - original draft, Writing - review and editing; Rob Phillips, Conceptualization, Resources, Data curation, Formal analysis, Supervision, Funding acquisition, Validation, Investigation, Methodology, Writing - original draft, Writing - review and editing; Ron Milo, Conceptualization, Resources, Data curation, Formal analysis, Supervision, Funding acquisition, Validation, Investigation, Methodology, Writing - original draft, Project administration, Writing - review and editing

*Competing interests:* The authors declare that no competing interests exist.

## Funding

| Funder | Grant reference number | Author |
| --- | --- | --- |
| National Institutes of Health | 1R35 GM118043-01 (Maximizing Investigators Research Award) | Rob Phillips |
| Weizmann Institute of Science | Charles and Louise Gartner professorial chair | Ron Milo |
| The Azrieli Foundation | Azrieli Fellow | Yinon M Bar-On |

The funders had no role in study design, data collection and interpretation, or the decision to submit the work for publication.

**Decision letter and Author response**
Decision letter https://doi.org/10.7554/eLife.57309.sa1
Author response https://doi.org/10.7554/eLife.57309.sa2

## Additional files

### Supplementary files

• Supplementary file 1. SARS-CoV-2 (COVID-19) by the numbers. This is a larger version of *Figure 1*.

## Data availability

This article is a compilation of previously published data; no new data were generated in this study.

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
