## [Decision Letter]

**Comments from reviewer #1:**

This article summarizes a lot of knowledge on corona viruses into a digestible format.

I have one major comment - on the first page, the values marked with an asterisk (derived from SARS-CoV-1, TGEV or MHV coronaviruses data) need to be more clearly indicated. I would suggest not using the asterisk at all and putting which virus the knowledge derives from each time. It will be visually less appealing, but I think appropriate - especially for the Maintenance of antibody response: ≈2-3 years* - statistic as it could be incredibly misleading.

A few minor comments -

- In graphic: the role of TMPRSS2 in priming the spike might be mentioned alongside ACE2 in the first graphic:

https://www.sciencedirect.com/science/article/pii/S0092867420302294

- On the first page - eclipse period and replication cycle are not the same, as defined later, so it is confusing on the first page why the eclipse period in parenthesis is stated as equal to replication cycle. This language could be more precisely and consistently defined throughout.

- In question 6 the terminology of polyprotein and polypeptide are intermingled. It probably makes sense for it to stay polyprotein as a polypeptide is any protein.

-In question 7 it’s a little confusing that there are two measures of the rate of mutation and one of these is the mutation rate.

-Glossary - Maybe Virion and other virus specific terms should be defined if this is intended for a general audience.

-In more on section, Q3: Virions and not viruses may be more appropriate ‘Even though there is no “burst”, we can still estimate the average number of viruses a single infected cell will produce’.

- For a scientific review, question 4, “How can an N95 mask block SARS-CoV-2?” should have referenced NIOSH criteria / requirement to be considered a specific filtration quality of a mask. N95 does not simply mean it removes 95% of all particles that are at least 0.3 μm. Looking at the technical specifications will explain why N95 masks may work (and may not be the best, but good enough). There are other ratings that are involved when discussion focused on particle types and sizes -- including ASTM Level, BFE (Bacterial Filtration Efficiency), PFE (Particle Filtration Efficiency), VFE (Viral Filtration Efficiency). At least 1 report from National Institute for Occupational Safety and Health (Rengasamy et al., 2017: https://doi.org/10.1080/15459624.2016.1225157) studied how these ratings were determined based on the specified methods. Experimentally determined PFE rating (at 0.1 μm) for the N95 masks that were used were ~99.8%.

- I was surprised there were no numbers on the inflammatory response or the causes of death (heart failure, etc:see Zhou et al. 2020. The Lancet 395:1054–1062). Perhaps this data is not yet firm enough yet.

Overall this is a visually appealing and timely summary.

James Fraser - UCSF

**Comments from reviewer #2:**

The manuscript by Bar-On et al. will be an extremely useful visual source of information for researchers and the general scientific audience on the numbers pertinent to the current SARS-CoV-2 pandemic. Overall, the graphic is clear and the back-of-the-envelope calculations are useful and support or add to existing published data. The manuscript has been careful to include that many of the numbers are still being updated or are unknown at this time until more experiments are done, but this collation of sources and data will be a useful launchpad for researchers studying the virus as well as educators and a more general audience. Due to the urgent dissemination of this information during the current crisis, I have only suggested just a few short edits that will add clarity for readers in the text. As is, I think the manuscript is well supported by the citations listed, will be interesting and appropriate for the *eLife* readership and I support its publication.

Some brief editing notes that will add clarity for readers:

- In the first panel (blue headers), please make it more clear which statistics given reference only studies done on other viruses including SARS-CoV-1, TGEV, or MHV. Currently this is an asterisk with small note in the blue panel, but since this applies to the entire large figure, please make this more obvious by making the text larger or drawing attention to it in some way. Also, use consistent viral name references to provide clarity to the readers (SARS is written in top “Genome” panel, then in asterisk SARS-CoV-1 is used. It may be useful to include both the full name and commonly used name.

In the “What can we learn from the mutation rate of the virus?” section – sentence “For example, with a concentration of …” I don’t think this calculation is helpful. You say that in a single mL of sputum (implying one human sample) that every possible base-pair mutation would be represented. While this calculation is correct for 1mL at that concentration of RNA, it’s highly unlikely/impossible that a single patient would be infected with many strains of the same virus and I don’t think it makes any specific point here. Even changing the language to not suggest a human sample would be better, unless there is a source suggesting this could be the case.

In the “How stable and infectious is the virion on surfaces?” section. The last sentence current reads “From the basic reproductive number R 0 ≈2-4 we can infer an upper bound on the risk of infection from touching a surface recently touched by an infected person.” But then do not provide a calculation or any additional information and the paragraph ends. Delete this sentence or include the additional calculations/information.

Xavier Darzacq

**Comments from reviewer #3:**

**-** Regarding paragraph 2 - Worth clarifying that this needs to be modified for households of more than 1 person.

**-** Regarding paragraph 5 - Worth making this statement a bit more tentative or adding a ref or explanation for the claim.

Regarding the sentence that starts "Multiplying the mutation rate . . ."

- Please consider adding something along the following lines: "assuming no fitness effects for the virus."

- I was also confused by how we know the evolutionary rate – by assuming a date of introduction arrived at independently from the sequence data?

**-** Regarding the sentence that starts "Multiplying the mutation rate . . ."

Add “on average”.

---

## [Author Response]

[We repeat the reviewer comments here in italic, and include our responses in plain text].

**Comments from reviewer #1**- I have one major comment - on the first page, the values marked with an asterisk (derived from SARS-CoV-1, TGEV or MHV coronaviruses data) need to be more clearly indicated. I would suggest not using the asterisk at all and putting which virus the knowledge derives from each time. It will be visually less appealing, but I think appropriate - especially for the Maintenance of antibody response: ≈2-3 years* - statistic as it could be incredibly misleading.

We modified the figure to indicate textually which virus was used for each measurement.

- In graphic - the role of TMPRSS2 in priming the spike might be mentioned alongside ACE2 in the first graphic:https://www.sciencedirect.com/science/article/pii/S0092867420302294

Following the reviewer’s comment, we added a mention of the fact that the spike protein is primed by TMPRSS2 both to the figure and the “definitions” section.

- On the first page - eclipse period and replication cycle are not the same, as defined later, so it is confusing on the first page why the eclipse period in parenthesis is stated as equal to replication cycle. This language could be more precisely and consistently defined throughout.

We modified the figure to note the timescale as the eclipse period, and modified the definitions section to clarify the distinction between the eclipse period and the latent period.

- In question 6 the terminology of polyprotein and polypeptide are intermingled. It probably makes sense for it to stay polyprotein as a polypeptide is any protein.

Now fixed.

- In question 7 it’s a little confusing that there are two measures of the rate of mutation and one of these is the mutation rate.

Following the reviewer’s comment we modified the first sentence of the paragraph to clarify the phrasing.

- Glossary - Maybe Virion and other virus specific terms should be defined if this is intended for a general audience.

We added a definition for the following terms: Virion, Polyprotein, Nasopharynx, latent period, and interval of half-maximal infectiousness, as well as the names of the various virus strains.

- In more on section, Q3: Virions and not viruses may be more appropriate ‘Even though there is no “burst”, we can still estimate the average number of viruses a single infected cell will produce’.

Now corrected.

- For a scientific review, question 4, “How can an N95 mask block SARS-CoV-2?” should have referenced NIOSH criteria / requirement to be considered a specific filtration quality of a mask. N95 does not simply mean it removes 95% of all particles that are at least 0.3 μm. Looking at the technical specifications will explain why N95 masks may work (and may not be the best, but good enough). There are other ratings that are involved when discussion focused on particle types and sizes -- including ASTM Level, BFE (Bacterial Filtration Efficiency), PFE (Particle Filtration Efficiency), VFE (Viral Filtration Efficiency). At least 1 report from National Institute for Occupational Safety and Health (Rengasamy et al., 2017: https://doi.org/10.1080/15459624.2016.1225157) studied how these ratings were determined based on the specified methods. Experimentally determined PFE rating (at 0.1 μm) for the N95 masks that were used were ~99.8%.

We thank the reviewer for this highly relevant information. We updated the text, and added reference to the NIOSH criteria, and the reference provided on the PFE of N95 masks.

- I was surprised there were no numbers on the inflammatory response or the causes of death (heart failure, etc:see Zhou et al. 2020. The Lancet 395:1054–1062). Perhaps this data is not yet firm enough yet.

We added a reference for the prevalence of preexisting conditions in the section discussing the difference between the Case Fatality Rate and Infected Fatality Rate. After reviewing the literature, we believe further research is needed to solidify estimates on cytokine concentrations, and therefore have not included estimates regarding their concentration at this current version.

We modified the figure to indicate textually which virus was used for each measurement.

In the “What can we learn from the mutation rate of the virus?” section – sentence “For example, with a concentration of …” I don’t think this calculation is helpful. You say that in a single mL of sputum (implying one human sample) that every possible base-pair mutation would be represented. While this calculation is correct for 1mL at that concentration of RNA, it’s highly unlikely/impossible that a single patient would be infected with many strains of the same virus and I don’t think it makes any specific point here. Even changing the language to not suggest a human sample would be better, unless there is a source suggesting this could be the case.

Indeed we do not expect the patient to be infected with many strains. We updated the text of the vignette and calculation to make it clearer.

In the “How stable and infectious is the virion on surfaces?” section. The last sentence current reads “From the basic reproductive number R 0 ≈2-4 we can infer an upper bound on the risk of infection from touching a surface recently touched by an infected person.” But then do not provide a calculation or any additional information and the paragraph ends. Delete this sentence or include the additional calculations/information.

We revised the ending accordingly.

Comments from reviewer #3:- Regarding paragraph 2 - Worth clarifying that this needs to be modified for households of more than 1 person.

There might be some modification possible but we are unsure what it should be.

- Regarding paragraph 5 - Worth making this statement a bit more tentative or adding a ref or explanation for the claim.

We updated the statement and made it more tentative as suggested.

- Regarding the sentence that starts "Multiplying the mutation rate . . ."- Please consider adding something along the following lines: "assuming no fitness effects for the virus."- I was also confused by how we know the evolutionary rate – by assuming a date of introduction arrived at independently from the sequence data?

We updated the text as suggested. The inference of the evolutionary rate is detailed in the reference cited.

- Regarding the sentence that starts "Multiplying the mutation rate . . ."- Add “on average”.

This number is not actually an average, but an estimate from the reported peak RNA concentrations, which range from 10^6^-10^11^ RNAs/mL. We have updated the text to make it more clear how we arrive at this number.